# FASTVPINNS: A FAST, VERSATILE AND ROBUST VARIATIONAL PINNS FRAMEWORK FOR FORWARD AND INVERSE PROBLEMS IN SCIENCE

**Divij Ghose, Thivin Anandh & Sashikumaar Ganesan**
Department of Computational and Data Sciences,
Indian Institute of Science, Bangalore
Bengaluru, India
{divijghose, thivinanandh, sashi}@iisc.ac.in

## ABSTRACT

Variational physics-informed neural networks (VPINNs), with h and p-refinement, show promise over conventional PINNs. But existing frameworks are computationally inefficient and unable to deal with complex meshes. As such, VPINNs have had limited application when it comes to practical problems in science and engineering. In the present work, we propose a novel VPINNs framework, that achieves up to a 100x speed-up over SOTA codes. We demonstrate the flexibility of this framework by solving different forward and inverse problems on complex geometries, and by applying VPINNs to vector-valued partial differential equations.

## 1 INTRODUCTION

In recent years, PINNs have been a popular technique to solve forward and inverse problems related to differential equations(Haghighat et al., 2021; Mao et al., 2020). PINNs approximate the solution of a PDE using a deep neural network, which is optimized my minimizing the residual of the PDE (Raissi et al., 2019). Since their introduction, several variants of PINNs have been proposed, which exploit different architectures or numerics to obtain better solutions (Cuomo et al., 2022). One such extension uses the variational form of the differential equations and numerical integration to compute the loss (Kharazmi et al., 2019; Khodayi-Mehr & Zavlanos, 2020). While several works have demonstrated than variational PINNs (VPINNs) to be more accurate than vanilla PINNs, particularly with h and p-refinement (hp-VPINNS), the vast majority of VPINN applications to date have only been limited to scalar problems in small domains with limited flexibility in the type of geometry that can be used (Kharazmi et al., 2021; Yang & Foster, 2021; Radin et al., 2023). One reason for this is the poor computational efficiency of VPINNs, where most implementations calculate the total loss by looping through each cell. Moreover, the simple reference transformation used in existing implementations means that they can only be applied to geometries that can be decomposed into rectangular cells. As a result, solutions of coupled equations on complex meshes, for example the one shown in Fig. 1, are infeasible in state-of-the-art hp-VPINNs codes.

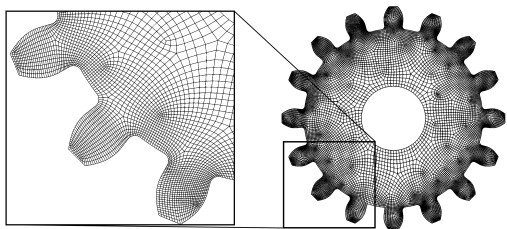

Figure 1: Mesh for a spur gear with 14,000 quad cells.

In this work, we propose a novel implementation of hp-VPINNs based on mapped finite elements (Wilbrandt et al., 2017), that vectorizes cell-based loops and replaces them with tensor calculations. This leads to considerable speed-ups by leveraging GPU computing, and lends itself to solutions on virtually any geometry that traditional techniques like the finite element method can handle.

## 2 METHODOLOGY

We present the mathematical preliminaries of both PINNs and hp-VPINNs in A. Assuming our domain is decomposed into $N_{\text{cells}}$ cells, each having $N_{\text{quad}}$ quadrature points and $N_{\text{test}}$ test functions, we compute the variational loss in a manner shown in Fig. 2a. We calculate the derivatives of the test functions, Jacobians and the quadrature weights once, since these do no change while training the network, and assemble them into a pre-multiplier matrix. However, when dealing with skewed cells in complex geometries, the Jacobian may vary at different quadrature points for each test function. By implementing a bilinear transformation to map the gradients of the test function from the reference cell to the actual cell, as shown in Fig. 2b, we can handle any complex geometry that can be decomposed into quadrilateral cells, without the need for a uniform mesh. Hence, by stacking the pre-multiplier matrices for each cell, we obtain a third-order tensor with dimensions $N_{\text{cells}} \times N_{\text{test}} \times N_{\text{quad}}$. We collect all the quadrature points from each cell and pass it through the neural network go, thereby obtaining the solutions at all points in a single forward pass, and the gradients of the solutions in a single backward pass. The solution gradients are assembled into a two-dimensional matrix of dimensions $N_{\text{quad}} \times N_{\text{cells}}$. This approach guarantees that the gradient computations for every cell across the domain are performed in a single backward pass through the neural network, instead of being repeated $N_{\text{cells}}$ times as in existing implementation of hp-VPINNs. A detailed description of the bilinear transformation can be found in B

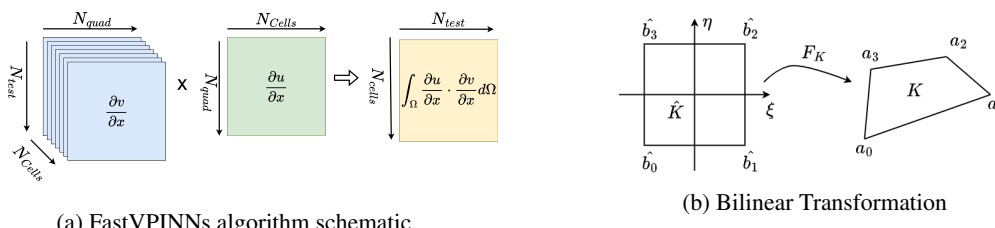

(a) FastVPINNs algorithm schematic

(b) Bilinear Transformation

Figure 2: **(a)** Tensor-based computation of the variational loss. **(b)** Bilinear transformations to handle reference transformations for quadrilateral cell.

## 3 RESULTS

In the following section, we examine the performance of FastVPINNs in terms of speed and accuracy. We use an NVIDIA RTX A6000 GPU with 48GB of device memory for training. The NVIDIA-Modulus library(NVIDIA Modulus) was used for PINNs results, while the code available on (Kharazmi, 2023) was used for obtaining a baseline performance for hp-VPINNs. For our test functions, we use Jacobi polynomials, denoted as $P_n$ for degree n, such that $P_n = P_{n+1} - P_{n-1}$.

### 3.1 FORWARD PROBLEMS WITH THE TWO-DIMENSIONAL POISSON'S EQUATION

We first demonstrate the performance of our code, with h and p-refinement, on forward problems by solving the two-dimensional Poisson equation on the unit square, with the given forcing function,

$$-\Delta u(x,y) = -2\omega^2 \sin(\omega x) \sin(\omega y) \quad (x,y) \in \Omega = [0,1]^2,$$
$$u(0,\cdot) = u(\cdot,0) = 0 \tag{1}$$

This problem has the exact solution

$$u(x,y) = -\sin(\omega x)\sin(\omega y) \tag{2}$$

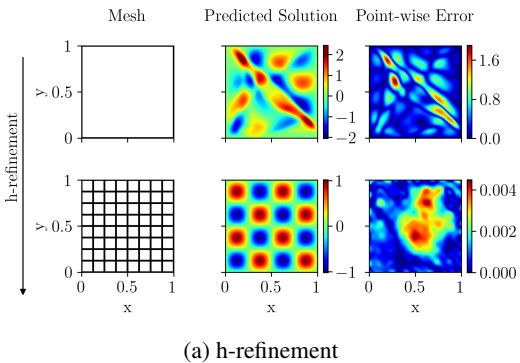 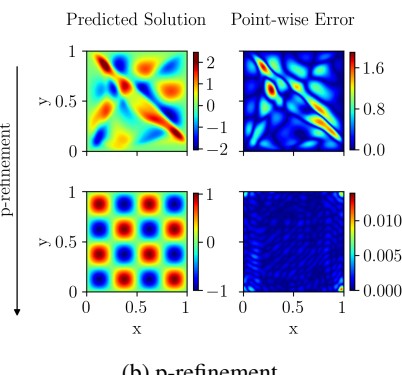

(a) h-refinement

(b) p-refinement

Figure 3: **(a)** Effect of h-refinement on accuracy. The first, second, and third columns are the domain decomposition, fastVPINN solution, and point-wise test error, respectively. From top to bottom: N_elem = 1, N_elem = 64. For each cell, we use 80 × 80 quadrature points and 5 test functions in each direction. **(b)** Effect of p-refinement on accuracy on 1 cell. The first and second columns are the fastVPINN solution and point-wise test error, respectively. From top to bottom: Ntest = 5 × 5 and Ntest = 20 × 20. The cell has 80 × 80 total quadrature points

For a test case with $\omega = 4\pi$, Fig. 3 shows that our framework exhibits improvement in the accuracy of the solution using h and p-refinement, as expected from any hp-VPINNs solver. We then illustrate the efficiency of our framework with that of PINNs (utilizing the Nvidia-Modulus Library, as shown in Modulus (2023) and hp-VPINNs (as shown in Kharazmi (2023)) in Fig. 4. Fig. 4a compares the median time per epoch required for solving Eq. 1 as the number of cells increases, with each cell having 25 quadrature points in total and 5 test functions in each direction. For a fair comparison, the PINNs solution is obtained using the same number of residual points, and we show our results for both FP32 and FP64 precision. We can see that the existing hp-VPINNs code scales linearly as the number of residual points increases, whereas the time required by our framework remains largely constant, offering up to a 100x speed-up. Further, in Fig. 4b, we show that our code consistently converges faster than PINNs, with the improvements being more noticeable as the solution frequency increases. We observe that our code is faster than PINNs, which could be because we skip the backward pass to compute second derivatives for calculating the loss.

## 3.2 Forward problems on a complex geometry with the two-dimensional convection-diffusion equation

To demonstrate the application of our framework to complex geometries, we solve the convection-diffusion equation given in Eq. 3 on the mesh shown in Fig. 1.

$$-\varepsilon\Delta u + \mathbf{b} \cdot \nabla u = f, \quad \mathbf{x} \in \Omega \tag{3}$$

where,

$$f = 50\sin x + \cos x; \quad \epsilon = 1; \quad \mathbf{b} = [0.1, \ 0]^T;$$

Fig. 5 shows that our code manages to achieve reasonable accuracy even on a complex mesh with a large number of cells, which was thus far infeasible for hp-VPINNs codes.

## 3.3 Forward problems with the two-dimensional incompressible Navier-Stokes equations

Since existing codes do not scale very well, as is evident from Fig. 4a, hp-VPINNs have not been applied to more complex problems in science that are of practical interest, like the incompressible Navier-Stokes equation. We present a novel application of hp-VPINNs to this problem, using our fastVPINNs framework, for the solution of the Kovasznay flow and lid-driven cavity flow. Fig. 6 shows that our framework can solve such coupled flow problems with relatively good accuracy.

To complete the discussion of our hp-VPINNs framework, we demonstrate that it is equally capable in solving inverse problems as shown in Section 3.4

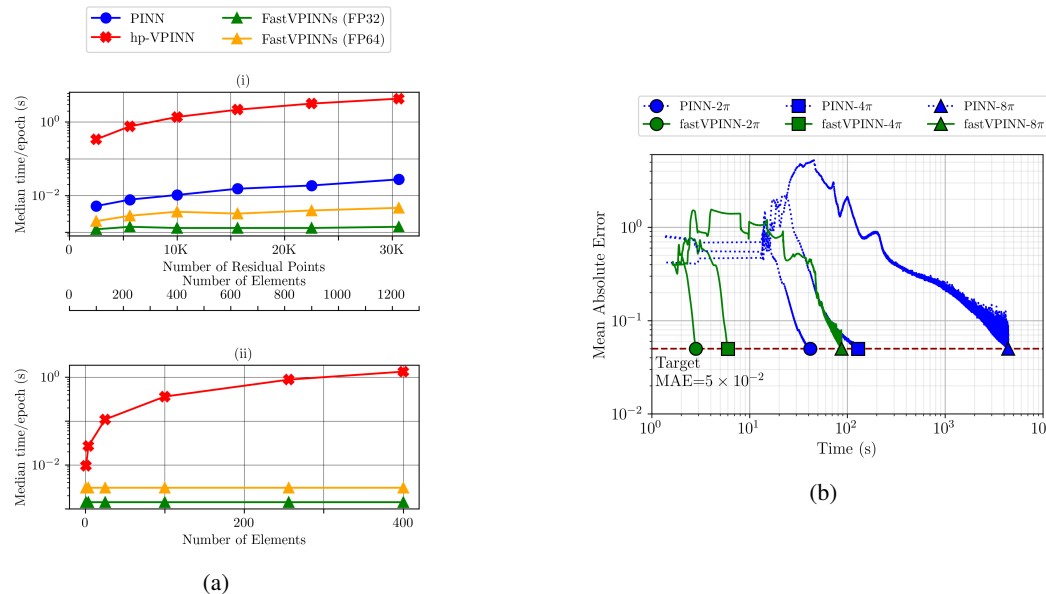

Figure 4: **(a)** (i) Variation of computational time with the number of quadrature (residual) points, plotted against the median time taken per epoch; (ii) Comparison of computational time between hp-VPINNs and fastVPINNs for varying numbers of cells. **(b)** Comparison between PINNs and fastVPINNS of total time taken to reach a target mean absolute error for different solution frequencies.

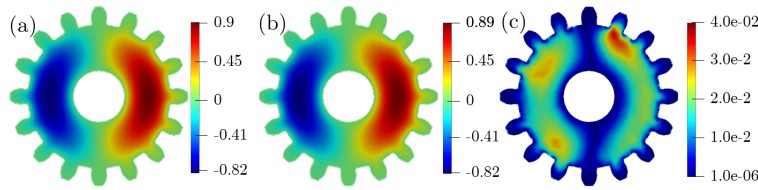

Figure 5: Solution of the convection-diffusion equation on a mesh for a spur gear with 14,192 quad cells. **(a)** Exact Solution. **(b)** Predicted Solution. **(c)** Point-wise error

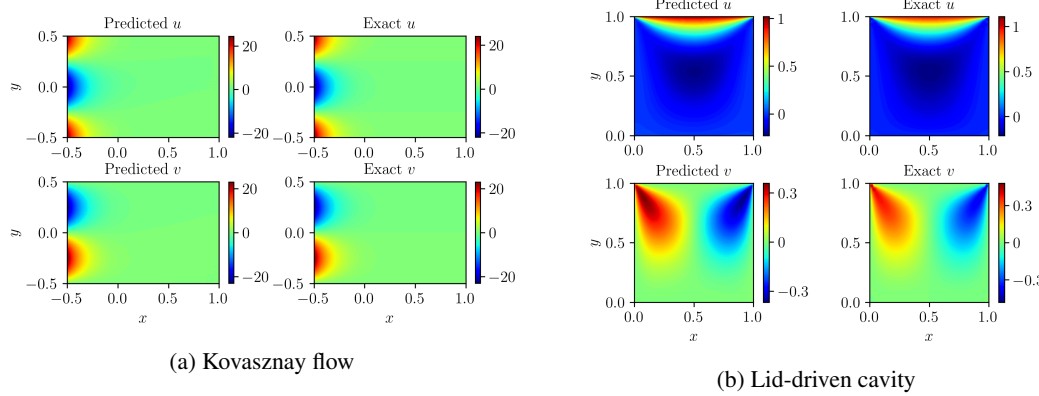

Figure 6: Preliminary solutions for the two-dimensional Navier-Stokes equations: **(a)** The Kovasznay flow for Re=40 and **(b)** Lid-driven cavity flow for Re=1 and lid velocity = 1.

### 3.4 INVERSE PROBLEMS: ESTIMATING THE DIFFUSION COEFFICIENT FOR THE TWO-DIMENSIONAL CONVECTION DIFFUSION PROBLEM

To complete the discussion of our framework, we show that it is equally capable in solving inverse problems on complex meshes. We add a sensor loss to our loss function by generating data from a finite cell solver for the problem shown in in Eq. 4.

$$-\left(\frac{\partial}{\partial x}\left(\epsilon(x,y)\frac{\partial u}{\partial x}\right) + \frac{\partial}{\partial y}\left(\epsilon(x,y)\frac{\partial u}{\partial y}\right)\right) + b_x\frac{\partial u}{\partial x} + b_y\frac{\partial u}{\partial y} = f, \quad \mathbf{x} \in \Omega \quad (4)$$

where,

$$f = 10; \quad \epsilon_{\text{actual}}(x,y) = 0.5\left(\sin x + \cos y\right); \quad b_x = 1.0; \quad b_y = 0.0;$$

We then predict the unknown spatially varying diffusion coefficient and the solution, as shown in Fig. 7.

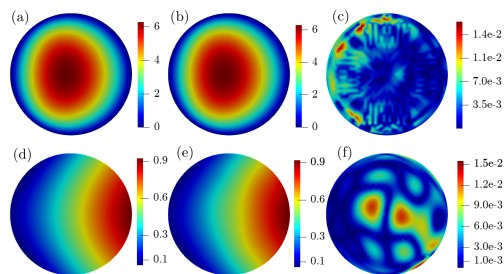

Figure 7: Exact and predicted solution and diffusion parameter for the inverse problem on a unit circle. **(a)** Exact solution($u$) obtained from FEM(ParMooN). **(b)** Predicted Solution FastVPINN. **(c)** Absolute error: FEM vs FastVPINN solution. **(d)** Exact diffusion parameter($\epsilon_{\text{actual}}$). **(e)** Predicted diffusion parameter by FastVPINN($\epsilon_{\text{predicted}}$) **(f)** Absolute error: $\epsilon_{\text{actual}}$ vs $\epsilon_{\text{predicted}}$

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

## A  PRELIMINARIES

### A.1  GOVERNING EQUATIONS

Consider a two-dimensional steady-state Poisson equation:

$$
\begin{aligned}
-\Delta u(x) &= f(x), &&\text{in } \Omega \subseteq \mathbb{R}^2, \\
u(x) &= g(x), &&\text{on } \partial\Omega.
\end{aligned}
\tag{5}
$$

Here, $x \in \Omega$, $\varepsilon$ represents the diffusion coefficient. In addition, $f(x)$ is a known source function with appropriate smoothness. The Dirichlet boundary condition $u(x) = g(x)$ is imposed on the domain boundary $\partial\Omega$.

### A.2  HP-VARIATIONAL PHYSICS INFORMED NEURAL NETWORK

In this section, we initially establish the variational form of the Poisson equation equation 5, followed by introducing the Variational Physics Informed Neural Network. Let $\mathrm{H}^1(\Omega)$ denote the conventional Sobolev space, and define

$$
V := \left\{ v \in \mathrm{H}^1(\Omega) : v = 0 \ \text{ on } \ \partial\Omega \right\}.
$$

The subsequent procedure consists of taking the equation equation 5, multiplying it by $v \in V$, integrating over $\Omega$, and then utilizing integration by parts on the second derivative term. For more detailed information, please refer to (Ganesan & Tobiska (2017)). Consequently, the variational representation of the Poisson equation can be formulated as:

Find $u \in V$ such that,

$$
a(u, v) = f(v) \quad \text{for all } v \in V,
$$

where

$$
a(u, v) := \int_\Omega \nabla u \cdot \nabla v \, dx, \qquad f(v) := \int_\Omega f v \, dx.
\tag{6}
$$

The domain $\Omega$ is then divided into an array of non-overlapping cells, labeled as $K_k$, where $k = 1, 2, \ldots, N_{\text{cells}}$, ensuring that the complete union $\bigcup_{k=1}^{N_{\text{cells}}} K_k = \Omega$ covers the entire domain $\Omega$. In this context, we define $V_h$ as a finite-dimensional subspace of $V$, spanned by the basis functions $\phi_h := \{\phi_j(x)\}$, $j = 1, 2, \ldots, N_{\text{test}}$, where $N_{\text{test}}$ indicates the total number of basis functions in $V_h$. As a result, the discretized variational formulation related to equation equation 6 can be written as follows,

Find $u_h \in V_h$ such that,

$$
a_h(u_h, v) = f_h(v) \quad \text{for all } v \in V_h,
\tag{7}
$$

where

$$
a_h(u_h, v) := \sum_{k=1}^{N_{\text{cells}}} \int_{K_k} \nabla u_h \cdot \nabla v \, dK, \qquad f_h(v) := \sum_{k=1}^{N_{\text{cells}}} \int_{K_k} f v \, dK.
$$

These integrals can be approximated by employing a quadrature rule, leading to

$$
\int_{K_k} \nabla u_h \cdot \nabla v \, dK \approx \sum_{q=1}^{N_{\text{quad}}} w_q \, \nabla u_h(x_q) \cdot \nabla v(x_q) \, ,
$$

$$
\int_{K_k} f v \, dK \approx \sum_{q=1}^{N_{\text{quad}}} w_q \, f(x_q) \, v(x_q) \, .
$$

Here, $N_{\text{quad}}$ is the number of quadrature points in a element.

The hp-Variational Physics Informed Neural Networks (hp-VPINNs) framework, as presented by Kharazmi et al. (2021), utilizes specific test functions $v_k$, where $k$ ranges from 1 to N_elem, that are localized and defined within individual non-overlapping element across the domain.

$$
v_k = \begin{cases} v^p \neq 0, & \text{over } K_k, \\ 0, & \text{elsewhere.} \end{cases}
$$

Here, $v^p$ represents a polynomial function of degree $p$. This selection of test and solution spaces results in a Petrov-Galerkin finite element method. Specifically, $u_h$ is estimated by $u_{\text{NN}}(x; W, b)$, which is the neural network solution, whereas the test function $v_h$ is a predetermined polynomial function. By utilizing these functions, we establish the element-wise residual of the variational form equation 7 with $u_{\text{NN}}(x; W, b)$ as

$$\mathcal{W}_k(x; W, b) = \int_{K_k} \left( \nabla u_{\text{NN}}(x; W, b) \cdot \nabla v_k \ - \ f \, v_k \right) dK.$$

$$\mathcal{B}(x; W, b) = u_{\text{NN}}(x; W, b) - g(x), \qquad \text{on } \partial\Omega.$$

(8)

Further, define the variational loss by

$$L_v(W, b) = \frac{1}{N_{\text{cells}}} \sum_{k=1}^{N_{\text{cells}}} |\mathcal{W}_k(x; W, b)|^2$$

$$L_b(W, b) = \frac{1}{N_D} \sum_{d=1}^{N_D} |\mathcal{B}(x; W, b)|^2 \, ,$$

and the cost function of the neural network in hp-VPINN as

$$L_{\text{VPINN}}(W, b) = L_v + \tau L_b.$$

(9)

Here, $L_b$ is the Dirichlet boundary loss as expressed in equation 8 and $\tau$ is a scaling factor applied to control the penalty on the boundary term.

## B    BILINEAR TRANSFORMATION

Let $b_0(-1, -1)$, $b_1(1, -1)$, $b_2(1, 1)$, $b_3(-1, 1)$ be the vertices of the reference element $\hat{K}$, see Figure 2b. For any function $u(X)$, we denote $u(X) = u(F_k(\hat{X})) = \hat{u}(\hat{X})$. Further, the derivatives of the function, $\partial u/\partial x$ and $\partial u/\partial y$ on the original element can be obtained in terms of the derivatives defined on the refernce element $\partial \hat{u}/\partial \xi$ and $\partial \hat{u}/\partial \eta$ as follows:

$$\hat{u}(\hat{X}) = \hat{u}(F_k^{-1}(X)) = u(X),$$

(10)

$$\frac{\partial \hat{u}}{\partial \xi} = \frac{\partial u}{\partial x}\frac{\partial x}{\partial \xi} + \frac{\partial u}{\partial y}\frac{\partial y}{\partial \xi},$$

(11)

$$\frac{\partial \hat{u}}{\partial \eta} = \frac{\partial u}{\partial x}\frac{\partial x}{\partial \eta} + \frac{\partial u}{\partial y}\frac{\partial y}{\partial \eta}.$$

(12)

we can express this relation as,

$$\begin{bmatrix} \frac{\partial \hat{u}}{\partial \xi} \\ \frac{\partial \hat{u}}{\partial \eta} \end{bmatrix} = \begin{bmatrix} (x_{c1} + x_{c3}\eta) & (y_{c1} + y_{c3}\eta) \\ (x_{c2} + x_{c3}\xi) & (y_{c2} + y_{c3}\xi) \end{bmatrix} \begin{bmatrix} \frac{\partial u}{\partial x} \\ \frac{\partial u}{\partial y} \end{bmatrix}.$$

where,

$$x_{c0} = \frac{(x_0 + x_1 + x_2 + x_3)}{4}, \qquad x_{c1} = \frac{(-x_0 + x_1 + x_2 - x_3)}{4},$$

$$x_{c2} = \frac{(-x_0 - x_1 + x_2 + x_3)}{4}, \qquad x_{c3} = \frac{(x_0 - x_1 + x_2 - x_3)}{4},$$

$$y_{c0} = \frac{(y_0 + y_1 + y_2 + y_3)}{4}, \qquad y_{c1} = \frac{(-y_0 + y_1 + y_2 - y_3)}{4},$$

$$y_{c2} = \frac{(-y_0 - y_1 + y_2 + y_3)}{4}, \qquad y_{c3} = \frac{(y_0 - y_1 + y_2 - y_3)}{4},$$

Finally, we have Finally, we have

$$\begin{bmatrix} \frac{\partial u}{\partial x} \\ \frac{\partial u}{\partial y} \end{bmatrix} = \frac{1}{D} \begin{bmatrix} (y_{c2} + y_{c3}\xi) & -(y_{c1} + y_{c3}\eta) \\ -(x_{c2} + x_{c3}\eta) & (x_{c1} + x_{c3}\xi) \end{bmatrix} \begin{bmatrix} \frac{\partial \hat{u}}{\partial \xi} \\ \frac{\partial \hat{u}}{\partial \eta} \end{bmatrix},$$

where, $D$ is the determinant of the Jacobian matrix

