# OpenReview forum: "FastVPINNs: A fast, versatile and robust Variational PINNs framework for forward and inverse problems in science"
_ICLR.cc/2024/Workshop/AI4DiffEqtnsInSci — AI4DiffEqtnsInSci @ ICLR 2024 Poster_

### Official Review · Reviewer_vEuC · 2024-02-26
**Authors propose Fast Variational physics-informed neural networks framework using tensor-based computation of variational loss and bilinear transformations for quadrilateral cell.**

**Rating:** 8
**Confidence:** 4

**Review:**

Authors propose FASTVPINN framework using tensor-based computation of variational loss and bilinear transformations for quadrilateral cell.

---

### Official Review · Reviewer_GTvP · 2024-02-26
**Review of fast VPINNs paper**

**Rating:** 5
**Confidence:** 3

**Review:**

This paper improves the efficiency of training hp-VPINNs. This is achieved by vectorizing computations over the hp-VPINN’s cells when computing the hp-VPINN loss function, allowing tensorized GPU operations to be used. The authors claim that this approach is orders of magnitude faster to train than PINNs and standard hp-VPINNs.

In general, reducing the training time of hp-VPINNs is important for them to be useful in realistic applications, and so the subject of this paper is well motivated. It does appear that the authors achieve considerable speedups compared to PINNs and standard hp-VPINNs.

There are some weaknesses that should be addressed:

-	This paper lacks a mathematical description of hp-VPINNs, their trial/test functions, and their loss function - such a description is essential to include. Furthermore, it is unclear whether the proposed approach uses a global trial function (like the original hp-VPINN paper) or local trial functions as well as local test functions – this should be clarified -  and the authors do not mentioned which test functions are used – this is essential to include.

-	It would be useful to have a mathematical explanation on how the bilinear transform allows vectorization, in the context of the hp-VPINN loss function.

-	The authors claim “the existing hp-VPINNs code scales exponentially as the number of residual points increases, whereas the time required by our framework remains largely constant”. This sounds unrealistic to me – what is the actual computational complexity of evaluating the hp-VPINN loss function and how does it scale with the number of residual points? This would be more convincing if stated mathematically (big O notation). Furthermore, I am unsure why the proposed approach is faster in terms of time/epoch than PINNs, because PINNs already use tensorized operations and are evaluated using the same number of residual points – more explanation would be useful.

-	In places the paper needs more proof-reading; please check for typos (e.g. Figure 4 I am not sure the subplot labels should be h-refinement and p-refinement, and in the introduction “where most implementations loop calculate the”)

In summary, the motivation is sound, but the paper lacks some essential implementation descriptions which limit its reliability in my opinion.

---

### Meta-Review · Area_Chair_RR3s · 2024-03-01

**Recommendation:** Accept (Poster)

**Metareview:**

The paper proposes a framework that speeds up training of hp-VPINNs for physics simulations by vectorizing, achieving considerable speedups over standard PINNs. While the motivation to improve efficiency is sound, the reviewers note limitations like insufficient mathematical details on the methods and some unclear implementation aspects that need to be addressed for the camera-ready version.

---

### Decision · Program_Chairs · 2024-03-02

Accept (Poster)